# Edible Insect Meals as Bioactive Ingredients in Sustainable Snack Bars

**DOI:** 10.3390/foods14040702

**Published:** 2025-02-18

**Authors:** Francesca Coppola, Silvia Jane Lombardi, Patrizio Tremonte

**Affiliations:** 1Institute of Food Science, Italian National Research Council, Via Roma 64, 83100 Avellino, Italy; 2Department of Agricultural Sciences, University of Naples “Federico II”, Piazza Carlo di Borbone 1, 80055 Portici, Italy; 3Department of Agricultural, Environmental and Food Sciences (DiAAA), University of Molise, Via De Sanctis snc, 86100 Campobasso, Italy; tremonte@unimol.it

**Keywords:** *Alphitobius diaperinus*, *Tenebrio molitor*, *Listeria*, *Escherichia coli*, *Penicillium expansum*, biopreservation, antimicrobial activity

## Abstract

Insect metabolites are known for their preservative potential, but the time-consuming and unsustainable extraction process compromises their transferability. This study aimed to identify user-friendly solutions based on the use of insect meals that could improve microbiological safety as well as consumer acceptability. In this regard, the antimicrobial activity of *Alphitobius diaperinus* and *Tenebrio molitor* meals against surrogate strains of Gram-positive (*Listeria monocytogenes*) and Gram-negative (*Escherichia coli*) pathogenic bacteria and mycotoxin-producing fungi (*Penicillium expansum*) was evaluated. Minimum inhibitory concentration values of between 3.12 mg/mL vs. *Listeria innocua* and 12.50 mg/mL vs. *Escherichia coli* were found. Based on this finding, a model food was developed also considering consumer acceptance. Statistical analysis of food preferences showed that nutritional and sustainability claims were the independent variables of greatest interest. Therefore, waste or by-products from other food chains were selected as co-ingredients for sustainability, nutritional, and sensory claims. Analysis of the chemical composition showed that the insect bar-style snack qualifies as a “high-protein” food, as protein provides more than 20% of the energy value. Based on the moisture (30%) and water activity (0.77) values, the bar could be classified as an intermediate-moisture food. The challenge test showed that the insect meal prevented the proliferation of intentionally added undesirable microorganisms. Conclusively, the findings complement the knowledge on the antimicrobial activities of insect meals, offering new possibilities for their use as natural preservative ingredients with nutritionally relevant properties.

## 1. Introduction

Human evolution, progress, and the growth of civilizations are closely linked to the progress and transition of agri-food systems [1,2]. Today, more than any other time, with more than 8 billion people having the right to healthy food, profound innovation is needed. The availability of food divides the planet into two parts: on the one hand, rich geographical areas with a surplus of food and a worrying obesity rate, and on the other, geographical areas where the right to food is denied to more than a billion people [3]. The handling is paradoxical: almost 40 percent of food, which requires high pressure on the earth to be produced, is wasted before it reaches consumers’ tables [4,5]. Moreover, these two 21st-century paradoxes are accompanied by a rise in antimicrobial resistance of food-borne pathogenic and/or spoilage microorganisms [6,7,8,9]. The advances and knowledge gained in the last decade in the field of food biotechnology and food protection offer various insights into innovations in the food system [10,11,12]. The use of plant- or insect-based proteins or bioactive substances as an alternative to chemical ingredients or conventional additives are the pillars on which the food system transition rests [13,14,15,16,17]. To date, four edible insects are permitted as feed and food ingredients in Europe: the lesser insect worm *Alphitobius diaperinus* (*A. diaperinus*) [18], the house cricket *Acheta domesticus* [19,20], the locust migratory worm *Locusta migratoria* [21], and the yellow worm *Tenebrio molitor* (*T. molitor*) [22,23]. Besides the ability to convert and enhance by-products of different origins [24,25], insects are also high in protein, healthy fats, vitamins, and minerals, characteristics that make them a promising resource to meet the growing global demand for food [26,27]. Moreover, their use in the food chain goes beyond direct consumption: products derived from insects, such as extract or purified bioactive compounds, offer new opportunities to improve food quality and safety [28]. The study of the antimicrobial properties of insects is receiving increasing attention. The first evidence of antimicrobial properties in insects was reported in the 1970s [29]. Certainly chitin, a linear polymer composed of N-acetyl-glucosamine that is particularly abundant in the adult and larval forms of insects, is responsible for the antimicrobial interest. The antimicrobial activity of chitin has already been extensively characterized in recent decades [30], as has the enhanced efficacy of chitosan. Chitosan shows higher efficacy due to positively charged amine groups interacting with bacterial cell surface components and higher solubility [31]. To date, extracts and peptides derived from insects have demonstrated effective antimicrobial activity against a wide range of pathogenic and spoilage microorganisms [32]. This has led to proposals for their use as natural preservatives in food [33]. Recent studies highlighted the potential of these compounds in controlling several bacteria, including pathogenic species [34,35,36] that extracted chitin from larvae and adults of *T. molitor* by deproteinization and demineralization, with yields of around 6% in larvae and 12% in adults. The same authors observed that chitosan, obtained by deacetylation of chitin, produced inhibitory activity against sporogenesis bacteria such as *Bacillus cereus* vs. Gram-positive ones such as vs. *Listeria monocytogenes* and *Staphylococcus aureus* and Gram-negative ones such as *Escherichia coli* [37]. Anti-fungal activity expressed by chitosan from insects was also found. Until a few years ago, conventional methods of extraction and subsequent de-methylation of chitin, using considerable and impactful amounts of solvents, acids, and chemical bases, had several environmental drawbacks. Only recently, several green extraction methods have also been successfully developed to extract chitin and chitosan from various resources, including insect sources. Overall, techniques based on the use of chemicals have been supplemented with physical operations, ranging from thermal action to the application of high-pressure CO_2_ via high-voltage pulsed electric fields [38,39]. Although green approaches for the extraction of bioactive compounds have been developed [40], the extraction requires technologically complex processes that could limit their large-scale adoption. In addition, besides chitin and chitosan, other antimicrobial compounds are present, such as medium-chain saturated fatty acids, with lauric acid (LA) in particular, and antimicrobial peptides (AMPs) [34,41,42,43]. Based on these considerations, insect meals represent an attractive alternative to purified compounds or extracts, offering an integrated and less elaborate approach to harnessing the antimicrobial benefits of insects. Moreover, with the presence of bioactive compounds, insect meals retain all the nutritional properties of the whole organism, making it a versatile ingredient from both functional and nutritional perspectives. In addition, the use of insect meals simplifies the production process, reducing costs and complexity, and ensures a more sustainable approach, minimizing waste and exploiting the full potential of raw materials. However, the achievement of the plethora of benefits associated with the use of insect meals is related to their antimicrobial efficacy and the extent to which it is comparable to that of purified compounds or extracts. To date, although insect meals have been proposed as an ingredient for food or feed, emphasizing their antimicrobial potential [36,41,44,45], no exhaustive studies are available on the antimicrobial efficacy of insects used directly as meal in food preparation. Gaps or unresolved issues are also those related to meeting consumers’ demands or resolving their hesitations. The factors determining their acceptability or rejection have been investigated through specific research and analyzed in reviews [46,47,48,49]. However, further investigation that considers consumer expectations, the bioactive potential of edible insects, and their use in food preparation in an integrated manner is necessary and remains a priority. Based on these considerations, the antimicrobial activity expressed by two edible insect meals against various microorganisms, including Gram-negative bacteria (*Escherichia coli*), Gram-positive bacteria (*Listeria innocua*), and fungi (*Penicilliium expansum*), was evaluated and compared to that produced by purified compounds such as chitin and chitosan or insect extracts. The possibility of using insect meal was not only related to its antimicrobial efficacy but also to the expectations and requirements of a heterogeneous sample of consumers. Finally, for the validation of the use of edible insects, the integrated consideration of several factors was adopted, based on specific challenge tests and consumer feedback.

## 2. Materials and Methods

### 2.1. Pre-Treatment Edible Insect

*A. diaperinus* and *T. molitor* larvae were chosen as edible insects. Both larvae were purchased from an insect company (Fauna Topics Zoobedarf, breeding and trade LLC, Marbach am Neckar, Germany) and grown in a controlled rearing facility according to European food standards. They are fed exclusively with plant-based feeds, such as mixtures of cereal flour and bran, without the use of antibiotics, hormones, and pesticides, according to GMP+ FC (Good Manufacturing Practice Feed Certification system). Twenty-four hours prior to the technological scalding process, the larvae were not fed to allow the gut to empty, and the room temperature was lowered to promote hibernation. They were then freeze-dried, ground, and sieved to produce fine powder and stored at room temperature until use. The final weight of the larvae was 1/3 of the original live weight.

#### Antimicrobial Activity and Minimum Inhibitory Concentration

Meals from the two insects were evaluated for their antimicrobial activity against *Escherichia coli* ATCC 8739 (as a surrogate of enterohemorrhagic *Escherichia coli*) [50], *Listeria innocua* ATCC 33090 (as a surrogate of *Listeria monocytogenes*) [51], and *Penicilliium expansum* ATCC 36200. All strains were purchased from the ATCC—American Type Culture Collection. Strains of *Escherichia coli* (*E. coli*), *Listeria innocua* (*L. innocua*), and *Penicilliu expansum* (*P. expansum*) were revitalized at 37 °C in Luria–Bertani broth (LB—Oxoid, Milan, Italy), at 28 °C in brain–heart infusion (BHI—Oxoid, Milan, Italy), and at 28 °C in potato dextrose broth (PD—Oxoid, Milan, Italy), respectively. The insect meals were suspended in sterile distilled water at a ratio of 1:3 *w*/*v*, and their inhibitory action was assessed by the agar well diffusion assay as previously described [52,53]. Briefly, 1 mL (inoculum concentration of 4.0 log CFU/mL) of each bacterial suspension was inoculated in 19 mL of proper soft media (0.7% agar), gently mixed, and poured onto Petri plates (BHI for *L. innocua*, LB for *E. coli*, and PD for *P. expansum*). On each plate, 7 mm-diameter wells were set up, within which 70 µL of meal suspension were added. Tetracycline disk 30 µg (Sigma-Aldrich, Milan, Italy) and sterile distilled H_2_O (70 µL) were used as negative and positive control, respectively. Plates inoculated with *P. expansum* were incubated at 28 °C for 24 h, while plates containing *L. innocua* or *E. coli* were incubated at 37 °C for 24 h. Antimicrobial activity was assessed by measuring the diameter of the halo formed around the wells after 24 h. Based on the halo diameter, the inhibitory activity was described as low (halo ≤ 10 mm), moderate (10 < halo < 20 mm), or high (halo > 20 mm). Five independent replicates were performed. In addition, the minimum inhibitory concentration (MIC) was determined as described in the final EUCAST 3.1 document [54] with some modifications [14]. Specifically, each larval meal suspension was added to the different culture media to obtain a concentration of between 0.2 and 4% (*w*/*v*) and placed in Petri dishes. The surface of the plates was inoculated with the different microbial strains at a concentration of 4.0 log CFU/mL and incubated at the respective temperatures. At the end of the incubation period, the plates were evaluated for the presence or absence of growth. MIC values were recorded as the lowest concentration of natural extract that completely inhibited growth. Each test was performed in triplicate. Finally, the effect of meal suspensions on the kinetic parameters of the three strains was evaluated: *E. coli*, *L. innocua*, and *P. expansum* were inoculated (6 log CFU/mL) in phosphate-buffered saline (PBS) in the presence of *T. molitor* meal or *A. diaperinus* meal suspension at the MIC value, and the growth kinetics were assessed using the structured dynamic model of Baranyi and Roberts [55]. DMFit software (v2.0) was used for this purpose. Kinetic parameters were analyzed to determine the antimicrobial score, arbitrarily defined as follows:$$Antimicrobial\_score=-\frac{logN\_end-logN\_0}{logN\_0}$$
where:
N_end = number of cells at the end of the observation period;N_0 = number of cells at the beginning of the observation period.

### 2.2. Food Design

#### 2.2.1. Consumer Survey—Assessment of Consumer Attitudes

Participants were recruited online by distributing a questionnaire to consumers from EU countries over a three-week period via the University of Molise’s e-mail networks, as well as among the research team’s social networks, and respondents were asked to share the survey link to increase coverage. To have a representative population of the different age groups, consumers aged 16 to 85 years were included. In addition, up to 10 consumers per year of age were accepted, with 700 interviewees collected (Appendix A). To this end, an age-related question was asked upstream of the questionnaire, which allowed respondents to proceed only if they were within the age range of interest. All respondents provided informed consent (Document S1) and, in the case of respondents under the age of 18, also parental consent (Document S2). Data were collected and processed anonymously, without identification of participants, and analyzed in aggregate form. The survey consisted of three parts:-The first contained demographic questions regarding age, gender, level of education, and place of residence (city above or below 50,000 inhabitants or rural villages).-The second part of the survey included questions aimed at ascertaining knowledge of the possibility of eating insects and edible insect-based foods.-The third part included questions aimed at understanding the propensity to consume insect-based foods and the characteristics (nutritional, environmental, sustainability) that should accompany such foods, as well as a specific question aimed at understanding the type of insect-based food most preferred.

The interviewees answered each question with a numerical value according to a hedonistic scale from 1 to 10.

#### 2.2.2. Analysis of Factors Influencing Consumer Attitudes

Data were collected using Google Forms and were analyzed to define a predictive model of the propensity to consume insects as a function of other variables, including respondents’ demographic characteristics (e.g., age, educational qualification, place of residence, etc.) as well as expected characteristics from insect foods (such as sustainability, nutritional claims). To this end, a multiple linear regression was applied using the R (v4.2.3) environment and RStudio software (v2022.07.0). 

### 2.3. Insect-Based Model Food: Preparation and Characterization

A bar-type snack was made using insect larvae meal and other ingredients chosen for their workability, nutritional, and sustainability characteristics.

#### 2.3.1. Ingredients and Bar-Snack Preparation

The ingredients used in the production of bars include edible insects, puffed grains and cereals, and waste and by-products of agro-food production. *T. molitor* larvae meal (8%) and *A. diaperinus* (8%) were used as edible insects. Marron glacé waste 8% (GMF Oliviero Ltd., Avellino, Italy), carob gum 6% (CioKarrua Ltd., Ragusa, Italy), and whey 26% (Mediterranea Biotecnologia Ltd., Termoli, Italy) were chosen for their sustainability potential as waste or by-products of other food processing. Puffed rice 4%, puffed quinoa 12%, flaxseed meal 8% (Prezzemolo & Vitale, Milan, Italy), chopped dark chocolate 6% (Barry Callebaut, Milan, Italy), and agave syrup 14% (Eridania, Bologna, Italy) were chosen for their sensory, nutritional, and technological properties. For the bar’s preparation (Figure 1), 80 g *T. molitor* larvae meal and 80 g *A. diaperinus* were combined with 80 g marron glacé waste, 40 g puffed rice, 120 g puffed quinoa, 80 g linseed meal, and 60 g chopped dark chocolate. The ingredients were thoroughly mixed and combined with 260 mL of whey in which 60 g of locust bean gum had previously been dissolved. Finally, 140 mL of agave syrup previously heated to 75 °C were added to the resulting mixture. The entire dough was shaped and cut to obtain 200 bars of approximately 50 g and a parallelepiped shape with a width of 2.5 cm, a height of 6 cm, and a depth of 0.6 cm.

#### 2.3.2. Chemical Composition

Moisture, carbohydrates, protein, fat, ash, and fiber contents were determined in bars according to the official methods of AOAC 2023 [56], specified below. In detail, the moisture of the burgers was determined based on 5 g of sample placed in an oven for 16 h at 105 °C, calculated as the difference between the weight of the fresh sample and the weight of the sample after drying and expressed as percentage of dry matter (d.m.) [56]. Protein content was determined using the Kjeldhal method [56]. Crude fat content was measured in accordance with AOAC method 960.39 using a Soxhlet apparatus [56]. Ash content was determined by heating the samples for 3 h at 550 °C. Dietary fiber content was determined using AOAC method 985.29 [56]; carbohydrates (C) were calculated by the difference as follows:C% = 100 − (M + P + L + A + F)
where M = moisture (%), P = protein content (%), L = lipid content (%), A = ash content (%), and F = dietary fiber (%).

#### 2.3.3. Physical–Chemical and Microbiological Analyses

The pH was determined in three different points of two samples, using a pH meter with a Mettler Toledo spear probe (Novate Milanese, Italy). Water activity was measured in triplicate using a Water Activity Meter CR2 (AQUALAB Instrument, Washington, DC, USA) on two different samples of snack bars. Microbial groups were enumerated as follows: 10 g samples were decimal-diluted in a sterile solution of 0.1% peptone water, homogenized in a Stomacher 400 Lab Blender (Seward Medical, London, UK) for 3 min, and serially diluted in the same sterile solution. Total mesophilic bacteria (TMC), *Enterobacteriaceae*, *Listeria* spp., and fungi were detected after appropriate incubation in proper media and conditions. Briefly, TMC were enumerated on plate count agar (PCA) (Oxoid, Milan, Italy) after incubation at 30 °C for 72 h. *Enterobacteriaceae* were enumerated on violet red bile glucose agar (VRBGA) (Oxoid) after incubation at 37 °C for 24 h. Fungi were enumerated on rose bengal agar (RBA, Oxoid, Milan, Italy) after 48 h of incubation at 28 °C, and *Listeria* spp. were enumerated on Listeria selective Oxford agar base with modified Listeria selective supplement (Oxoid, Milan, Italy) after 24 and 48 h of incubation at 37 °C. Each experiment was carried out in duplicate, and results were expressed as the mean of measurements and standard error.

#### 2.3.4. Sensorial Characterization

The sensory evaluation of snack bars was conducted by 20 judges, aged between 22 and 65 years, specifically trained in the sensory analysis of snacks with a focus on bars. From 40 candidates, after 10 training sessions, a panel of 20 judges was formed with optimal uniformity of response. All respondents provided informed consent (Document S3). The panel of judges, led by a panel leader, conducted a threefold sensory analysis on the bar using a 9-point hedonic scale to assess appearance, color, aroma, flavor, texture, stickiness, aftertaste, and overall liking. The panelists considered the specific evaluation criteria for each attribute (Appendix A) as described below:-Appearance, an attribute describing the overall look of the bar, including shape, surface perception, presence of visible insect fragments, and uniformity, was evaluated according to the following criteria: Does the bar look attractive and appetizing? Is the shape consistent and well modeled? Are visible insect components present and do they affect acceptability?-Color, aimed at describing the appropriateness and attractiveness of the color in relation to consumer expectations, was expressed as a summative judgment with respect to the following criteria: Is the color uniform and natural? Does the color correspond to the expectations of a protein/energy bar? Are there any discolorations or unattractive shades?-Aroma was reported as a summative judgement with respect to the following questions: Is the aroma inviting or neutral? Are there strong or unpleasant aromas related to insects? Is the aroma consistent with that of conventional protein bars?-Flavor, which expresses the overall taste profile, including sweetness, bitterness, umami, and potential off flavors, was reported as the summative result of the following evaluation criteria: Is the taste balanced and pleasant? Are there any off flavors? Does the bar have a desirable level of taste?-Texture, understood as mouthfeel and structural integrity of the bar (hardness, chewability, and softness) is expressed as a summative judgement of the following criteria: Is the texture soft, firm, or too hard? Does it crumble, does it stick to the teeth, or is it grainy? Is chewability acceptable?-Stickiness, the degree of adherence of the bar to the fingers and to the oral cavity during consumption, is expressed as a summative judgement of the following criteria: Is it excessively sticky when handled? Does the bar leave a sticky residue in the mouth? Does the level of stickiness affect overall acceptability?-Aftertaste, which identifies the residual taste that remains in the mouth after swallowing, is expressed as a summative judgment of the following criteria: Is it pleasant, neutral or undesirable? Is it long lasting? Does the aftertaste encourage or discourage further consumption?-Overall liking, that is, the overall acceptability of the bar based on all sensory attributes combined, is expressed as a summative judgment of the following criteria: Would the consumer consume the bar again? Is the bar as enjoyable as or better than conventional protein bars?

### 2.4. Antimicrobial Validation of Insect Larvae Meal

The antimicrobial effect of the two insects’ larvae meal was evaluated by means of in situ tests by setting up an appropriate challenge test to simulate multiple microbial contamination conditions. For this purpose, the insect-based bars were inoculated with 4 log CFU/g of a microbial cocktail consisting of a mixture of three microbial strains referring to *L. innocua* ATCC 33090, *E. coli* ATCC 8739, and *P. expansum* ATCC 36200. Prior to use, the microbial cultures were revitalized in nutrient broth (Oxoid, Milan, Italy) and, in the exponential phase, the strains were inoculated into the mixture for challenge testing. The development of *Listeria* spp., *Escherichia* spp., and *Penicillium* spp. in the bars was evaluated during the storage period (30 days).

The data were compared with those found in bars prepared with the same ingredients but without insects and inoculated equally with the same mixture of microorganisms.

### 2.5. Statistical Analyses

The data on the inhibition halo and antimicrobial score against the three target microorganisms were subjected to analysis of variance (ANOVA) followed by a Bonferroni’s post hoc test to identify differences and their levels of significance (*p* < 0.05, *p* < 0.01, or *p* < 0.001) between conditions. The influence of independent variables on the propensity to consume was assessed by means of multiple linear regression analysis. The data obtained from the chemical, sensory analysis, and challenge tests were subjected to a *t*-test to identify the differences and their significance levels (*p* < 0.05, *p* < 0.01, or *p* < 0.001) between the bars prepared without (control batch) and with insect meals (insect-based bar batch).

## 3. Results and Discussion

### 3.1. Meal Alphitobius diaperinus and Tenebrio molitor Larvae: Antimicrobial Activity

To assess the antimicrobial effectiveness expressed by edible insect larvae, used as nutritionally relevant ingredients in food preparation, their anti-*Listeria*, anti-*Escherichia*, and anti-*Penicillium* spectrum of action was evaluated. Specifically, the effect of meals from the two insects on the growth of *E. coli* ATCC 8739, *L. innocua* ATCC 33090, and *P. expansum* ATCC 36200 was investigated and is shown in Figure 2. Insect larvae meal in all tests significantly interfered with the growth of the three microorganisms, resulting in clear inhibition halos around the wells. The meal used, in suspension in DMSO at 10% (*w*/*v*), produced inhibition halos against the target microorganisms, ranging from 30 mm (vs. *E. coli* ATCC 8739) as the lowest value to the highest value of 45 mm against *L. innocua* ATCC 33090. The results obtained from the different meals by two insects did not show substantial or significant differences (*p* > 0.1) between them in terms of their effects on the target microorganisms. On the contrary, significant differences depended on the sensitivity of the different strains to the insect meals. *L. innocua* ATCC 33090 showed the highest sensitivity, resulting in significantly higher sensitivity to both *P. expansum* ATCC 36200 (*p* < 0.01) and *E. coli* ATCC 8739 (*p* < 0.001). In addition, the latter was the least sensitive strain, showing a significant difference (*p* < 0.05) even compared to *P. expansum* ATCC 36200.

In all cases, the inhibition halos were significantly higher than those produced by antibiotic disks, namely, gentamycin disk (against *E. coli* and *L. innocua*) and cycloheximide disk (against *P. expansum*) used at a concentration of 10 mg/mL and 30 mg/mL, respectively. In fact, the diameter of the halos produced by both gentamicin and cycloheximide never exceeded 20 mm. These preliminary findings from the evaluation of inhibition halos are further confirmed by the relative MIC data (Table 1).

Both insect meals exhibited inhibiting activity, with MIC values of 3.12 mg/mL towards *L. innocua* ATCC 33090 and 6.25 mg/mL towards *P. expansum* ATCC 36200, while the highest MIC value of 12.5 mg/mL was observed towards *E. coli* ATCC 8739. Therefore, the meal showed an inhibitory capacity against target microorganisms when used at concentrations of between 0.6% and 1.5%. Considering that insect meals, as pointed out by several authors [57,58,59], can be used in food preparation at concentrations of 10% and even 50%, our results on antimicrobial efficacy are relevant for the biopreservation-conscious food industry. The remarkable antimicrobial activity of insect meals used as a raw material without any treatment is a novelty in scientific literature and in food industry research and development. To date, scientific literature includes numerous studies on the bioactive substances of edible insects and their antimicrobial potential [35,60]. However, the investigation of antimicrobial action has mainly focused on molecules extracted from edible insects, evaluating the efficacy of pure substances, their derivatives, or complex extracts [61,62,63]. Our results show that the antimicrobial action of insect meals is comparable or superior to that of insect complex extracts or pure insect compounds such as chitin. As can be seen from Figure 3, the antimicrobial effect expressed by the larvae meal of *A. diaperinus* and *T. molitor* in doses equal to MIC values is superior to both that produced by the protein extract of the larvae (*p* < 0.05) and that produced by pure chitin (*p* < 0.001) used in amounts compatible with MIC values [62,64,65].

Thus, the antimicrobial activity of insect larvae meals can be attributed to various bioactive molecules. Further studies will be necessary to identify and quantify the contribution of each bioactive substance. However, it can now be assumed that meals, without further treatment, can easily be used as an ingredient to counteract the development of specific microbial species, such as *L. innocua*, *E. coli*, or *P. expansum*, which are feared not only in high-moisture but also in low-moisture foods [66,67,68].

### 3.2. Use of Edible Insects as Antimicrobial Ingredients: Food Design

It is widely recognized that consumer opinion plays a crucial role in the design of novel foods [69,70]. Understanding consumer attitudes cannot possibly be neglected if the innovation intends to introduce much-discussed ingredients, such as edible insects [71]. For this reason, responses from a large sample of consumers (700 consumers evenly distributed over an age range of 16 to 85 years) were collected via a questionnaire designed to investigate familiarity with edible insects, propensity to purchase or consume insect products, and factors that might influence the acceptability of any insect-based food product. The population of respondents is representative of different age groups, diverse levels of education, and different places of residence (Appendix A). Specifically, the interviewees are equally distributed between male and female genders and equally distributed according to the size of their cities of residence. On the contrary, the majority stated that they had a university degree in terms of qualification. The survey showed that most consumers interviewed were aware of the possibility of using insects in food preparations. In fact, out of the total number of respondents, only 7% did not know that insects, whole or as meal, could be used as a food ingredient. Different attitudes emerge regarding the propensity to consume insects. The analysis of the answers shows that place of residence and level of education are not factors influencing propensity to purchase. On the contrary, a relationship was found between age and propensity to purchase. The predictive model of the propensity to consume (Figure 4), based on multiple regression analysis, highlighted a relationship between age and propensity to consume.

A bell-shaped (Gaussian) curve with a smaller slope on the right side than on the left describes the relationship between the propensity to consume insect-based foods and the age of the interviewees. Thus, the youngest age groups, 16–18 years, and adults over 70 years declared a lower propensity to consume insect-based foods. The 35–40 age group contained the consumers who declared the highest propensity to consume these foods. These findings are in line with evidence found in other European regions, according to which older individuals show greater resistance to incorporating insects into their diet than younger ones [71,72]. The predictive model of the propensity to consume insect foods also revealed which factors most influence consumer attitudes: sustainability characteristics and nutritional claims are the factors that showed a relationship with propensity to consume, also as a function of age. Specifically, a positive regression between sustainability traits and propensity to purchase was found in the younger consumer groups (age < 45 years). In the age groups of over 45 years, sustainability traits were found to be indifferent without influencing the propensity to consume. On the contrary, in these age groups (over 45 years), nutritional claims regarding insect-based foods positively influenced the choice. Based on consumer attitudes and the predictive model of consumer propensity, an insect-based food model was defined for the validation of the antimicrobial action and acceptability of *A. diaperinus* and *T. molitor*. Considering that the bar-type snack is the preferred staple food with insects (Figure 5), a bar-type snack was developed to meet specific nutritional and sustainability claims.

To this end, ingredients were selected that not only satisfy the intake of specific nutrients but also meet the aspects of the circular economy by reusing by-products and food waste from different food supply chains. The environmental pressure of food industry waste and by-products and their possible reuse has been widely studied and is still a topic of cutting-edge research [73,74,75,76]. Based on these acquisitions, among the ingredients to be used in the formulation of bar-type snacks, waste from confectionery processing (marron glacé waste); milk-processing waste or by-products, such as whey [77]; as well as waste from carob processing, such as carob gum [14], were selected.

### 3.3. Insect-Based Bar-Type Snack: Chemical, Nutritional, Microbiological, and Sensorial Features

The choice of ingredients and their use have resulted in a product for which 50% of the raw materials used are by-products, waste, or co-products of other food processes. If the use of insects is added to this percentage, about 60% of the product is obtained from sustainable or low-impact sources. Sustainability claims, as also highlighted by a recent review [78], could remove some consumer distrust and increase the propensity to consume atypical foods, such as insect-based foods. Considering the data acquired through our survey, the sustainability claim would increase the propensity to consume, especially among younger age groups. Noteworthy are the benefits related to macronutrient composition provided by the special choice of raw materials combined with the use of insect meals (Table 2). The use of ingredients such as carob bean gum, shifted linseed meal, and agave syrup, together with insect larvae meal, allows the bar to meet the requirements of Regulation (EC) [79] to qualify as a “protein source” food. In addition, the use of insects provides a significant protein contribution that allows the bar to qualify as a “high-protein” food [79], as more than 20% of the energy value of the food is provided by protein.

Achieving these characteristics in composition and, consequently, in nutritional claims makes it possible to meet the expectations of specific consumer groups, increasing their propensity to consume insect-based food products. The combination and concentration of the different ingredients also influenced the moisture content and the chemical–physical characteristics of the bars. The samples of the batch of bars prepared with the use of insects and those of the control batch without insects were characterized by the same moisture and water activity (aw) values: 30% and 0.776, respectively. These values are characteristic of intermediate-moisture food (IMF), a product category that is gaining more and more attention, as it has very similar characteristics to fresh food products, but with a longer shelf life [80]. However, the scientific literature in recent years has highlighted evidence and concerns regarding the microbiological safety of low-to-intermediate-moisture food products [81,82,83]. In addition, the pH of our samples is not a limiting factor for microbial survival or proliferation, being around 6.3. Samples from both batches immediately after packaging showed microbial-load levels of no concern: total mesophilic bacteria never exceeded 1000 CFU/g, yeast load levels were around 1000 CFU/g, and both *Listeria* spp. and *E. coli* were undetectable. No significant differences between the samples of the two batches (with or without insect meal) were found. Although microbiological-count levels are not of concern, several authors have pointed out that diverse food-borne illnesses are associated with low-moisture foods [81]. In the development of a new food, it is widely recognized that the assessment of sensory quality is crucial feedback to validate innovative solutions [84,85,86]. Sensory acceptance gains crucial importance in the case of the use of insects, which, despite being part of the diets of more than 100 countries in Asia, Africa, and South America, are still poorly accepted in the usual diet [57]. Novel foods based on the use of edible insects, if adapted to Western sensory preferences and food aesthetics, could be more appealing to consumers. Therefore, incorporation into bar-type snacks could bode well for better acceptance. As is shown in Figure 6, consumer acceptance was medium to high, with no significant differences compared to the control. Other authors [87] have pointed out that the practice of incorporating insects into familiar products without them being easily recognizable as insects can help overcome neophobia and increase acceptability. In terms of color, the use of insects also produced no significant differences (*p* > 0.05). This result can be attributed to the use of other ingredients, which, having a similar color to insect meal, helped camouflage the presence of insects. Parameters ranging from texture to flavor and aroma through stickiness could be influenced by using insects.

Some authors [88,89] have shown that the use of up to 15 percent insect powder does not significantly affect the sensory quality of the products. In the present study, the insect larvae meal of the two insects, used to the extent of 16%, produced a positive effect on texture and stickiness. No significant differences from the control were found in aroma, flavor, or the sensory attribute “aftertaste”.

### 3.4. Antimicrobial Validation

The effectiveness of anti-*Listeria*, anti-*E. coli*, and anti-*Penicillium* action was evaluated by performing a specific challenge test. The load trends of the three microorganisms in the insect-based bar intentionally inoculated with a multi-strain cocktail referring to the three species were compared with those found in control samples inculcated with the strain cocktail but without having been fortified with insect meal. Significant differences (*p* < 0.05) as a function of meal use were found by studying the kinetic parameters of the microbial populations. Indeed, as previously reported by other authors [90], the study and comparison of growth rate is a useful approach to evaluate the effect of specific treatments. In our study, the growth rate (Figure 7) for all microorganisms analyzed in the control samples was positive and showed an increase in loading over time. In contrast, negative growth-rate values were found for the microorganisms analyzed in the insect samples, indicating a decrease during the storage time.

Furthermore, significant differences (*p* < 0.05) were found between the growth or growth rates of the different microorganisms in both the conventional and the insect-based bars. In contrast to what was observed in vitro, *Escherichia* spp. was the most sensitive (*p* < 0.05) compared to both *Listeria* spp. and the presumed *Penicillium* spp. These findings could be due to the higher resistance of *Listeria* and *Penicillium* to environmental factors. Indeed, it is recognized that *Listeria* and fungi are the microorganisms most feared in products with low or intermediate moisture [81,91]. In contrast, in the model food, *Escherichia*, although less sensitive to insect meals in vitro than *Listeria*, was affected not only by the inhibiting effect of the meal but also by environmental factors, particularly moisture and aw. Overall, two fundamental aspects emerge from the results obtained: in products with intermediate moisture such as snack bars, possible pathogenic contaminants such as *Listeria*, *Escherichia*, and *Penicillium* can develop, and insect meals allow for an effective reduction in microorganisms. Based on these results, insect meals not only represent an important ingredient for nutritional claims but could also serve as a bio-preservative ingredient. Furthermore, these results fill critical knowledge gaps on the antimicrobial properties of insect meals and complement previous research, which has largely focused on compounds derived or extracted from insects. Although purified extracts such as chitin, chitosan, and antimicrobial peptides (AMPs) are recognized for their antimicrobial activity and proposed as potential food additives, extraction processes are often resource-intensive and have environmental impacts [35,36]. Insect meal, on the other hand, offers a more practical, sustainable, and effective alternative, as it maintains the nutritional properties of the whole organism and simplifies the production process. Therefore, insect meals as bio-preservatives represent an interesting innovation in the food industry. At the same time, however, they raise safety concerns and the need to establish a new regulatory framework, as they may pose a risk to allergy sufferers [92]. Since insect products are considered “novel food”, they require a preliminary scientific assessment by the EFSA. In this regard, it should be noted that the use of *A. diaperinus* and *T. molitor* meals in food preparations is properly authorized and regulated [19,20,23].

## 4. Conclusions

The findings of this study provide a valuable foundation for the use of insect meals as a multifunctional ingredient with both nutritional and bio-preservative properties. Notably, sensory analysis and consumer surveys highlighted the broad acceptance of insect meals, demonstrating their potential for incorporation into food products. In addition, Insect meals have demonstrated their antimicrobial efficacy against surrogate strains of *L. monocytogenes*, *P. expansum* and *E. coli* in intermediate moisture food models, contributing new evidence to the growing body of research on the control of microorganisms in low-moisture foods, a difficult area where conventional antimicrobial strategies often fail. In conclusion, this study improves the current knowledge on the antimicrobial properties of insect meals, demonstrating their viability as a natural and sustainable antimicrobial ingredient for food protection. By addressing both microbial safety and consumer expectations, the results align with ongoing efforts to develop innovative and sustainable solutions to agribusiness and food safety challenges. These findings further underscore the potential of insect meals as a practical alternative to conventional additives, offering the food industry new opportunities for innovation in line with nutritional, safety, and sustainability priorities.

## Figures and Tables

**Figure 1 foods-14-00702-f001:**
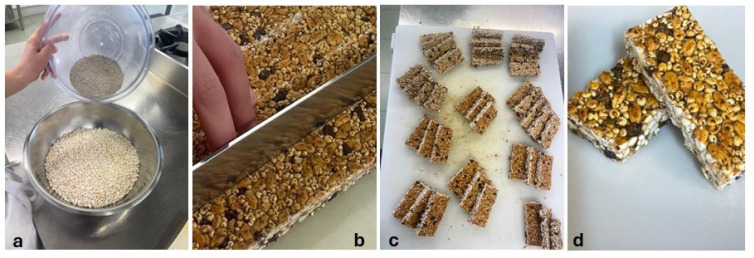
Sequence of photos relating to snack-bar preparation: (**a**) insect meals (in upper bowl) and other ingredients (in lower bowl); (**b**) cutting of formed and shaped dough; (**c**,**d**) finished product.

**Figure 2 foods-14-00702-f002:**
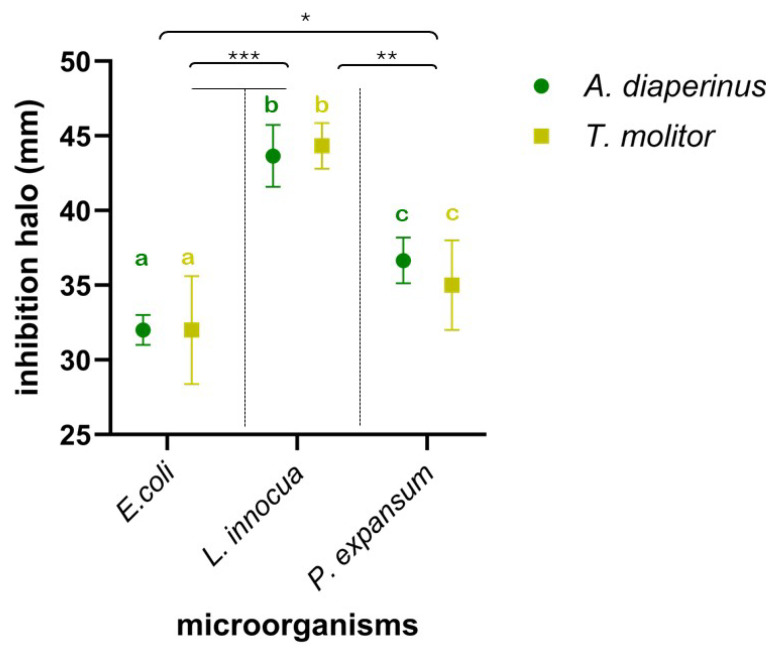
Inhibition halo plot (mm) of *Alphitobius diaperinus* (green) and *Tenebrio molitor* (yellow) meals against *Escherichia coli* ATCC 8739, *Listeria innocua* ATCC 33090, and *Penicilliium expansum* ATCC 36200. Different letters indicate a significant difference between microorganisms based on the statistical ANOVA test. Asterisks indicate significant difference values (*, *p* < 0.05; **, *p* < 0.01; ***, *p*< 0.001) in attributes among the 3 microorganisms.

**Figure 3 foods-14-00702-f003:**
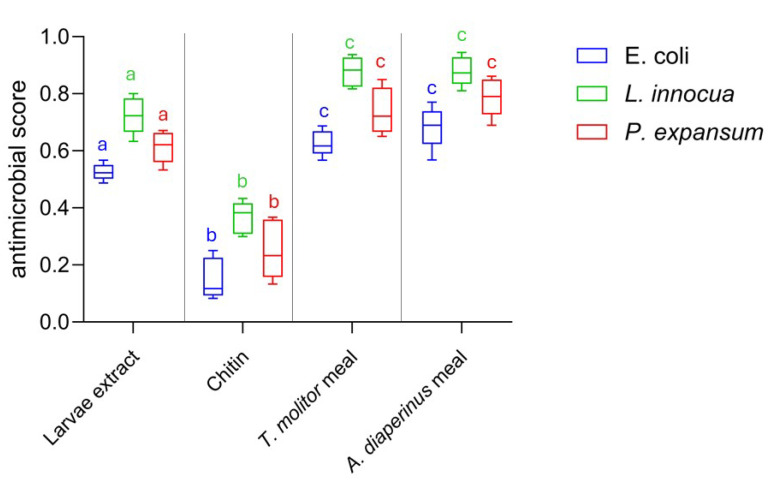
Box-and-whisker plots showing the antimicrobial score produced by *A. diaperinus* meal, *T. molitor* meal, chitin, and larvae extract against *Escherichia coli* ATCC 8739 (blue), *L. innocua* ATCC 33090 (green), and *P. expansum* ATCC 36200 (red). Different letters with the same color indicate significance difference between microorganisms depending on different treatments (larvae extract, chitin, insect meals) based on the statistical ANOVA test.

**Figure 4 foods-14-00702-f004:**
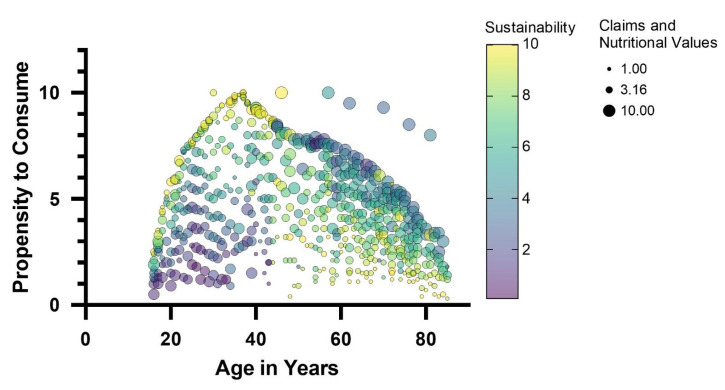
Predictive model of the propensity to consume insect-based foods by multiple linear regression of several variables, including consumers age as well as sustainability and nutritional claims. The size of the bubbles is directly related to the expectation of nutritional claims; the color of the bubbles (purple, yellow) is related to the expectation of nutritional claims on a scale of 1 to 10.

**Figure 5 foods-14-00702-f005:**
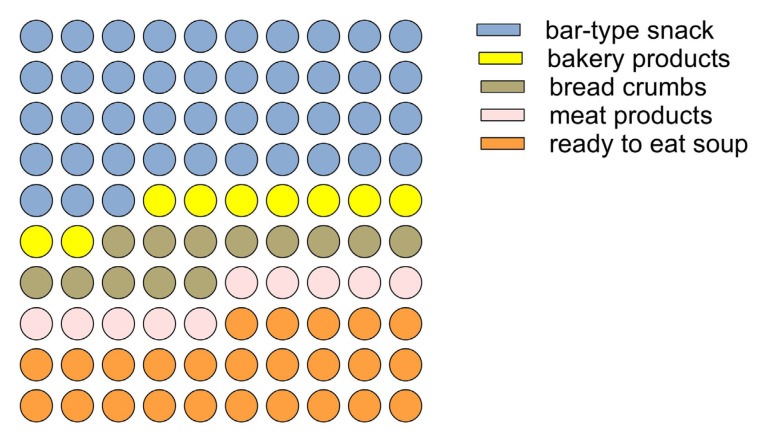
Part of the whole graph shows the preference for insect-based food types.

**Figure 6 foods-14-00702-f006:**
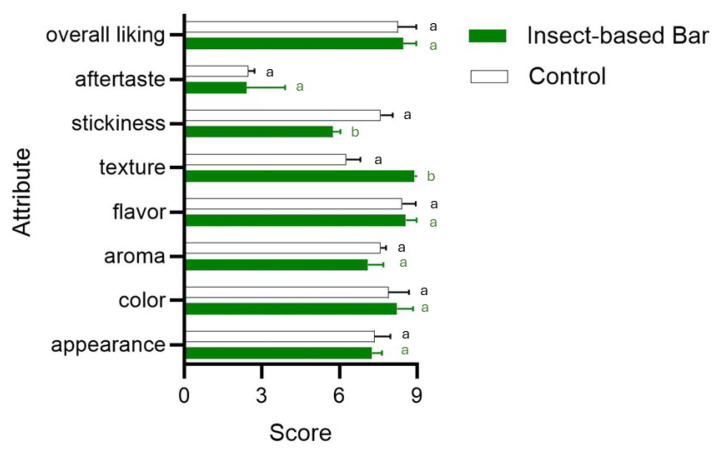
Bar plot showing the acceptability level (9-point hedonic scale) of sensory attributes in samples from conventional (control) and insect-based snack bars. Different letters indicate significant differences in attributes between the batches. Statistical tests were carried out using a *t*-test.

**Figure 7 foods-14-00702-f007:**
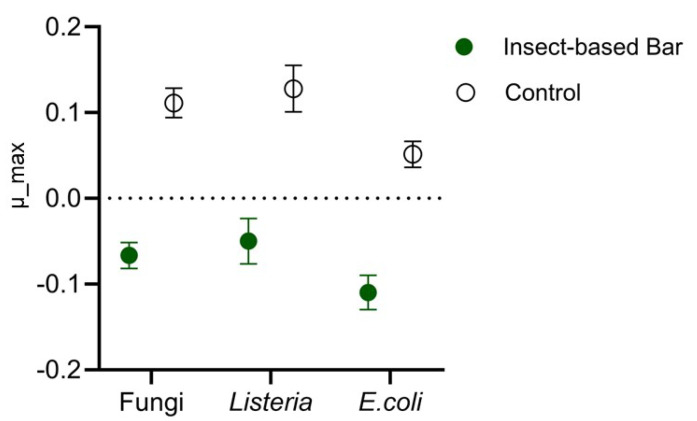
Maximum growth rate (m_max) of fungi (presumably *Penicillium*), *Listeria*, and *Escherichia* in insect-based or conventional (control) snack bars intentionally inoculated with a pathogen surrogate microorganism cocktail.

**Table 1 foods-14-00702-t001:** Minimum inhibitory concentration (MIC) values (mg/mL) of *Alphitobius diaperinus* and *Tenebrio molitor* meals against *Escherichia coli* ATCC 8739, *Listeria innocua* ATCC 33090, and *Penicilliium expansum* ATCC 36200.

Parameters	*Alphitobius diaperinus* Meal	*Tenebrio molitor* Meal
*E. coli* ATCC 8739	12.50	12.50
*L. innocua* ATCC 33090	3.12	3.12
*P. expansum* ATCC 36200	6.25	6.25

**Table 2 foods-14-00702-t002:** Chemical composition of the conventional snack bar (control) and the snack bar with the insect meal (insect-based bar).

Parameters	Insect-Based Bar	Control Bar
Moisture (%)	30.57 (±0.31) ^a^	30.51 (±0.28) ^a^
Protein (%/d.m.)	23.85 (±0.42) ^a^	10.86 (±0.23) ^b^
Lipid (%/d.m.)	11.3 (±0.29) ^a^	8.31 (±0.33) ^b^
Ash (%/d.m.)	2.12 (±0.11) ^a^	2.01 (±0.16) ^a^
Carbohydrates (%/d.m.)	48.21 (±0.51) ^a^	63.84 (±0.37) ^b^
Dietary fiber (%/d.m.)	14.52 (±0.38) ^a^	14.98 (±0.28) ^a^
Kcal from protein (%)	23.53 (±1.01) ^a^	10.91 (±0.96) ^1^
Kcal	302.65 (±2.98) ^a^	298.65 (±3.01) ^a^

^1^ Means within a row with different letters are significantly different (*p* < 0.001) based on statistical *t*-test.

## Data Availability

The original contributions presented in this study are included in the article/Appendix A. Further inquiries can be directed to the corresponding author.

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
