# Peer review of "Edible Insect Meals as Bioactive Ingredients in Sustainable Snack Bars"

_foods, 2025, doi:10.3390/foods14040702_

Round 1
Reviewer 1 Report
Comments and Suggestions for Authors
The study provides a novel perspective on using whole insect meals instead of isolated extracts, addressing sustainability and practicality concerns in food preservation. This approach simplifies the production process and aligns with eco-friendly food innovation trends. The study proposes a food model considering sustainability and nutritional claims, which is a significant step towards practical application. However, further details on the formulation and sensory evaluation process would be valuable for understanding how the claims were validated.
Please see my detail comments:
Abstract: please provide more detail information about obtained results
Introduction: sufficient, provide most important information based on up to date references
Methods: presented in detail. All points are clear and easy to understand.
Results and discussions: all points are presented in detail. My questions are:
Were there any variations in antimicrobial efficacy between Alphitobius diaperinus and Tenebrio molitor meals? If so, what factors could explain these differences?
Did you analyze the chemical composition of the insect meals to identify specific bioactive compounds responsible for antimicrobial activity?
Are there any regulatory challenges in approving insect meals as bio-preservatives, especially concerning food safety and allergenicity?
Author Response
Dear reviewer, thank you for taking the time to review the manuscript and for your suggestions. The responses (R) to each comment (C) are listed below:
General Comment: The study provides a novel perspective on using whole insect meals instead of isolated extracts, addressing sustainability and practicality concerns in food preservation. This approach simplifies the production process and aligns with eco-friendly food innovation trends. The study proposes a food model considering sustainability and nutritional claims, which is a significant step towards practical application. However, further details on the formulation and sensory evaluation process would be valuable for understanding how the claims were validated.
General Response: The auditor's suggestions were respected and the formulation of the bar, sensory analysis and other activities (statistical analysis, significant differences between the conditions adopted, etc.) were more detailed in accordance with the recommendations of the auditor.
C1: Abstract: please provide more detail information about obtained results
R1: As suggested, more detailed information on the results has been provided in the abstract. Variations are highlighted in yellow
C2: Introduction: sufficient, provide most important information based on up to date references
R2: The introduction has been revised. Some sentences have been summarized with the most salient and up-to-date information.
C3: Were there any variations in antimicrobial efficacy between Alphitobius diaperinus and Tenebrio molitor meals? If so, what factors could explain these differences?
R3: The effects of the two meals do not differ significantly. This concept was also further detailed in the discussion of the results (section 3.1).
Did you analyze the chemical composition of the insect meals to identify specific bioactive compounds responsible for antimicrobial activity?
R4: The antimicrobial activity of flours can be traced back to the presence of chitin and antimicrobial peptides. This was reported in the discussion section. Therefore, in future research perspectives attention could be paid to the different bioactive components of flours and their contribution in protection.
C5. Are there any regulatory challenges in approving insect meals as bio-preservatives, especially concerning food safety and allergenicity?
R5. I thank the reviewer for the pertinent consideration. The topic was addressed in the final part of the discussion, highlighting how insect meal can be considered an interesting innovation in the food industry. At the same time, however, they raise safety issues and the need to clearly establish a new regulatory framework. Since insect products are considered ‘Novel Food’, they require a preliminary scientific assessment by EFSA. In this regard, it should be noted that the use of meals of Alphitobius diaperinus and Tenebrio molitor in food preparations is properly authorized and regulated.
Reviewer 2 Report
Comments and Suggestions for Authors
Insects represent a significant novel food resource. This manuscript evaluates the antimicrobial activity of two edible insects (Alphitobius diaperinus and Tenebrio molitor) and the characteristics of the developed snack bar, while also investigating consumer attitudes. The manuscript presents a comprehensive study; however, it lacks depth and requires enhanced logical coherence. The statistical results need to be described with greater precision. Here are some specific comments:
1. Include specific data descriptions in the abstract.
2. Line 291 should refer to L. innocua instead of L innocua.
3. Lines 292 and 296, in section 2.5 Statistical analyses, should specify the levels of significance testing. Given the multiple significance levels (0.1, 0.05, 0.01, 0.001), all results related to significance in the manuscript should indicate the testing level, as seen in Line 294. The different letters in Figure 1 should denote which significance level they represent.
4. It is recommended to provide photographs of the insect-based food products.
5. Table 1 should include the units for the indicators.
6. In Figure 2, E. coli should be italicized.
7. The specific criteria for sensory evaluation should be uploaded.
8. In Line 355, the authors did not provide Table S1. The specific methods for this section of the study should be detailed in section 2. Materials and Methods.
9. In Line 464, clarify how the 16% addition level was determined.
10. The conclusion section is overly lengthy and needs to be condensed; some discussion points should not be included in the conclusion.
Author Response
Dear reviewer, thank you for taking the time to review the manuscript and for your suggestions. The Responses (R) to each comment (C) are listed below:
General comment. Insects represent a significant novel food resource. This manuscript evaluates the antimicrobial activity of two edible insects (Alphitobius diaperinus and Tenebrio molitor) and the characteristics of the developed snack bar, while also investigating consumer attitudes. The manuscript presents a comprehensive study; however, it lacks depth and requires enhanced logical coherence. The statistical results need to be described with greater precision. Here are some specific comments:
General Response.
The reviewers' comments were very helpful and made it possible to expand the different sections of the manuscript. The abstract has been implemented; methods, such as those related to sensory analysis, have been better described; the results and discussion section has been revised by clarifying significant differences between the different samples; finally, the conclusions have been condensed.
C1. Include specific data descriptions in the abstract.
R1. The abstract has been supplemented by reporting the main results and highlighting the data
C2. Line 291 should refer to L. innocua instead of L innocua.
R2. The correction was done
C3. Lines 292 and 296, in section 2.5 Statistical analyses, should specify the levels of significance testing. Given the multiple significance levels (0.1, 0.05, 0.01, 0.001), all results related to significance in the manuscript should indicate the testing level, as seen in Line 294. The different letters in Figure 1 should denote which significance level they represent.
R3. As reported by the reviewer, there were inconsistencies in the version of the paper regarding the significance levels of the differences found. The suggestions were accepted, inconsistencies were removed and the tests used were better described in section 2.5.
C4. It is recommended to provide photographs of the insect-based food products.
R4. The materials and methods section has been supplemented with a sequence of images relating to the bar preparation, from the insects to the finished product via the mixture of other ingredients that compose the bar.
C5. Table 1 should include the units for the indicators.
R5. Done
C6. In Figure 2, E. coli should be italicized.
R6. Done
C7. The specific criteria for sensory evaluation insect-based bar using a 9-point hedonic scale to assess appearance, color, aroma, flavor, texture, stickiness, aftertaste and overall liking.
R7. The materials and methods section were supplemented with specific criteria for the sensory evaluation of insect bars. In addition, the sensory analysis form has been included as supplementary material.
C8. In Line 355, the authors did not provide Table S1. The specific methods for this section of the study should be detailed in section 2. Materials and Methods.
R8. The sections (Material and Methods as well as Results) were revised by reporting how the respondents were recruited, how the data were collected and managed, and by attaching the informed consent form as supplementary material.
C9. In Line 464, clarify how the 16% addition level was determined.
R9. Done
- The conclusion section is overly lengthy and needs to be condensed; some discussion points should not be included in the conclusion.
R10. Done
Round 2
Reviewer 2 Report
Comments and Suggestions for Authors
The author has responded to the comments and carefully revised the manuscript.